# Matriptase-2 and Hemojuvelin in Hepcidin Regulation: In Vivo Immunoblot Studies in *Mask* Mice

**DOI:** 10.3390/ijms22052650

**Published:** 2021-03-06

**Authors:** Jan Krijt, Jana Frýdlová, Iuliia Gurieva, Petr Přikryl, Martin Báječný, Andrea U. Steinbicker, Martin Vokurka, Jaroslav Truksa

**Affiliations:** 1Institute of Pathophysiology, First Faculty of Medicine, 128 53 Prague, Czech Republic; jkri@lf1.cuni.cz (J.K.); jana.frydlova@lf1.cuni.cz (J.F.); gurieva.jul@seznam.cz (I.G.); pprik@lf1.cuni.cz (P.P.); mbaje@lf1.cuni.cz (M.B.); mvoku@lf1.cuni.cz (M.V.); 2Department of Anesthesiology, Intensive Care and Pain Medicine, University Hospital Muenster, University of Muenster, 48149 Muenster, Germany; andrea.steinbicker@ukmuenster.de; 3Institute of Biotechnology, Biotechnology and Biomedicine Center of the Academy of Sciences and Charles University in Vestec, Czech Academy of Sciences, 252 50 Vestec, Czech Republic

**Keywords:** hepcidin, transferrin receptor, neogenin, Hjv, Tmprss6, Tfrc, Tfr2

## Abstract

Matriptase-2, a serine protease expressed in hepatocytes, is a negative regulator of hepcidin expression. The purpose of the study was to investigate the interaction of matriptase-2 with hemojuvelin protein in vivo. Mice lacking the matriptase-2 proteolytic activity (*mask* mice) display decreased content of hemojuvelin protein. Vice versa, the absence of hemojuvelin results in decreased liver content of matriptase-2, indicating that the two proteins interact. To further characterize the role of matriptase-2, we investigated iron metabolism in *mask* mice fed experimental diets. Administration of iron-enriched diet increased liver iron stores as well as hepcidin expression. Treatment of iron-overloaded *mask* mice with erythropoietin increased hemoglobin and hematocrit, indicating that the response to erythropoietin is intact in *mask* mice. Feeding of an iron-deficient diet to *mask* mice significantly increased spleen weight as well as the splenic content of erythroferrone and transferrin receptor proteins, indicating stress erythropoiesis. Liver hepcidin expression was decreased; expression of *Id1* was not changed. Overall, the results suggest a complex interaction between matriptase-2 and hemojuvelin, and demonstrate that hepcidin can to some extent be regulated even in the absence of matriptase-2 proteolytic activity.

## 1. Introduction

Absorption of iron from the diet is regulated by the hepatocyte-derived peptide hepcidin; the regulation of hepcidin expression represents one of the most extensively studied aspects of iron metabolism. During the past two decades, it has become increasingly clear that the regulation of hepcidin synthesis depends on a number of proteins located at the hepatocyte plasma membrane. Homozygous mutations in genes encoding HFE, transferrin receptor 2 and hemojuvelin (HJV) can result decreased hepcidin expression and the development of hemochromatosis, indicating that these proteins act as positive regulators of hepcidin synthesis. Rather unexpectedly, a new and potent negative regulator of hepcidin expression was identified in 2008. The matriptase-2 (MT2) protein, encoded by the *TMPRSS6* gene, has originally been described already in 2002 [1], but its exact function was unknown at that time. It encodes a transmembrane serine protease expressed almost exclusively in the hepatocyte. Patients with homozygous *TMPRSS6* mutations display microcytic anemia associated with inappropriately high urinary hepcidin [2]. Similarly, mice with *Tmprss6* mutations display high hepatic hepcidin (*Hamp*) mRNA content and severe microcytic anemia [3,4].

The elucidation of the function of matriptase-2 in hepcidin regulation has been a subject of several studies, which have not produced definitive results so far. It has been unequivocally established that in the absence of MT2 the bone morphogenetic protein (BMP) signaling pathway is hyper-activated, as evidenced by reports on increased *Id1* mRNA content and liver phosphorylated SMAD protein content in *Tmprss6*-deficient mice [5,6]. It is not exactly known how this overactivation of the BMP/SMAD signaling is achieved. The very first paper examining the relationship between matriptase-2 and BMP signaling clearly demonstrated that in vitro matriptase-2 cleaves the hemojuvelin protein [7], an essential component of BMP signaling pathway [8]. Based on this finding, it has been widely accepted that the same mechanism functions also in vivo. The MT2/HJV interaction has further received strong support from the observation that mice lacking both MT2 activity and HJV protein show the same phenotype as mice deficient in HJV only [9]–the interpretation of these data is that the loss of MT2 activity cannot be expected to affect iron metabolism if its substrate, HJV, is absent. The proposed cleavage of HJV by MT2 moreover suggests the possible existence of an elegant regulation mechanism, in which the activity of MT2 regulates hepcidin expression by influencing the actual amount of hemojuvelin at the hepatocyte plasma membrane.

In 2011, the search for the physiological function of MT2 has become even more interesting by the observation that the administration of erythropoietin (EPO) fails to correct the microcytic anemia present in *Tmprss6*-mutated mice [10]. In wild type mice, EPO not only increases the production of red blood cells, but also dramatically decreases hepcidin expression [11,12]. Conversely, in *Tmprss6*-mutated mice, red blood cell parameters as well as liver hepcidin mRNA content are unaffected by EPO administration [6,10,13]. This observation led to the conclusion that functional MT2 is essential for hepcidin downregulation by accelerated erythropoiesis [6]; however, the exact role of MT2 in erythropoiesis-mediated hepcidin downregulation has not yet been defined. Shortly after the discovery of erythroferrone (ERFE) as the erythropoietic regulator of hepcidin expression [14] it has been hypothesized that functional MT2 could be essential for the correct function of ERFE [15]. The most obvious function of MT2 could in this respect be the proteolytic processing of ERFE or its putative [16] hepatocyte receptor; nevertheless, at present such a mechanism appears unlikely as ERFE probably does not need a hepatocyte receptor for its function [17].

An alternative look at the role of matriptase-2 in hepcidin regulation has emerged during the last few years. First, it has been postulated that MT2 cleaves not only HJV, but a whole array of iron-metabolism related proteins at the hepatocyte plasma membrane [18]. More recently, it has been proposed that the proteolytic activity of matriptase-2 is dispensable for the regulation of hepatocyte hepcidin [19], leading to the interpretation that proteolytic cutting is not the key to MT2 activity [20]. At present, the mode of action of matriptase-2 in hepcidin regulation is still not completely clear.

The purpose of the current report is to summarize the in vivo results obtained on the interaction between MT2 and HJV, and to present new data on hepcidin regulation by iron overload and deficiency in *Tmprss6*-mutated mice. The results strongly support interaction between HJV and MT2 proteins, but also demonstrate that some degree of hepcidin gene regulation is retained in mice lacking MT2-dependent proteolytic activity. The question whether in vivo MT2 cleaves other potential BMP/SMAD targets such as the endogenous BMP receptors remains at present unresolved.

## 2. Results

### 2.1. Absence of the Protease Domain of Matriptase-2 Results in Decreased Content of Hemojuvelin Protein

In our earlier study [21] we reported decreased liver HJV protein content in livers of *Tmprss6*-deficient mice obtained from the laboratory of Dr. Carlos López-Otín. This finding was unexpected, since the absence of HJV-cleaving enzyme would intuitively be expected to result in increased HJV protein content. We have subsequently confirmed our finding in *mask* mice [13], which lack the proteolytic domain of matriptase-2 [3], and whose phenotype is similar to *Tmprss6*−/− mice. Intriguingly, in direct contrast to our results, it has very recently been reported that *Tmprss6*−/− mice display increased liver HJV protein content [19]. This discrepancy can probably be explained by different antibodies, different processing of liver tissue as well as by different procedures used for antibody validation.

In mouse liver homogenates prepared at pH 8 in buffers containing a NP40-type detergent, HJV is detected by the R&D AF3634 antibody under reducing conditions as a pair of two bands of approximately 35 and 20 kDa [21]; the band at 35 kDa is stronger and can be easily quantified. Both bands very probably originate from cleavage of full-length mouse HJV (UniProt entry Q7TQ32) at the labile Asp–Pro bond at position 165 [22]. As can be seen in Figure 1A, in samples prepared from *mask* mouse livers the intensity of the 35 kDa band is significantly decreased. The blot also showed the absence of HJV-related signal in samples from *Hjv*−/− mice run as negative controls. Interestingly, when analyzing liver samples from mice injected with adeno-associated virus encoding *Hjv-Flag* under the control of a liver-specific promoter, strong new bands were detected at the position of the full-length HJV protein (Figure 1A). These results indicate that HJV expressed from the AAV2/8-*Hjv-Flag* vector might be processed differently from endogenous HJV, and suggest that, for in vivo studies, the use of knock-out animals as negative controls is the preferred method for antibody validation. 

### 2.2. Hemojuvelin-Deficient Mice Display Significantly Decreased Content of Full-Length Matriptase-2

Since there is strong evidence that MT2 and HJV proteins interact, it is conceivable that the absence of one protein at the hepatocyte plasma membrane could affect the expression of the other protein. Because the proportion of plasma membrane proteins in whole liver lysate is too low to allow detection of less abundant proteins such as MT2, we have previously attempted to enrich the liver homogenate in plasma membrane proteins. To this purpose, liver homogenates prepared without the addition of detergent were spun at 3000 g and the resulting pellets were subjected to extensive washing [23]. This procedure enabled the isolation of plasma membrane-enriched protein fraction in a reasonable yield (about 1 mg of protein per gram of liver) and allowed detection of MT2 protein by the Abcam ab56182 antibody. In agreement with the above hypothesis, we found significantly decreased full-length MT2 protein content in the liver plasma membrane-enriched fraction from HJV-deficient mice (Figure 2A). Interestingly, livers from HJV-deficient mice displayed, in addition to decreased MT2 protein content, a decrease in neogenin (NEO1) protein content (Figure 2B), apparently confirming the hypothesis that HJV, MT2 and NEO1 form a complex at the hepatocyte plasma membrane [24].

When evaluating MT2 immunoblots, it is important to note that, like other serine proteases, MT2 is expected to undergo activation by autocleavage [25]. Activation of MT2 is expected to result in two protein fragments, of which one is attached through its transmembrane domain to the plasma membrane, while the fragment containing the active site is tethered to the membrane-bound protein by disulfide bonds. Intriguingly, no bands corresponding to these two protein fragments have been so far detected in vivo, and all MT2 immunoblots reported only the detection of the full-length protein at approximately 125 kDa, as also seen in Figure 2A.

### 2.3. Mask Mice Upregulate Hepcidin When Fed an Iron-Enriched Diet

It is well known that the absence of functional MT2 protein results in hyper-activation of the liver BMP-SMAD signaling pathway and in hepcidin upregulation. Until very recently [26], it has not been investigated whether iron overload can further increase hepcidin in mice without functional MT2. Feeding of a diet containing excessive amount of iron (2% as carbonyl iron) to female *mask* mice for two months dramatically increased liver non-heme iron (from 43 ± 8 µg/g to 1424 ± 386 µg/g wet weight). Hepcidin mRNA content, as well as *Id1* and *Bmp6* content increased in iron-fed *mask* mice (Figure 3A). These results confirm the recently obtained data from *Tmprss6*-deficient mice injected with iron dextran [26], indicating that the regulation of hepcidin expression through BMP proteins is intact in mice lacking functional MT2.

### 2.4. Erythropoietin Activates Erythropoiesis in Iron-Pretreated Mask Mice

*Tmprss6*-deficient mice and *mask* mice have low iron stores and are therefore unable to increase hemoglobin levels or hematocrit following EPO administration [6,10,13]. Placing female *mask* mice on a diet containing 2% of carbonyl iron for 2 months increased liver iron content, while the blood count parameters normalized and exceeded the wild-type values (Figure 3B). These results demonstrate that the loss of MT2 activity affects erythropoiesis solely by limiting iron availability. EPO administration to iron-overloaded *mask* mice resulted in further increase in hemoglobin and hematocrit (Figure 3B). However, EPO administration to iron-overloaded *mask* mice did not decrease their high *Hamp* expression (Figure 3A). Neither did EPO administration influence the increased content of liver phosphorylated SMAD proteins observed in iron-treated *mask* mice (Figure 3C).

### 2.5. Mask Mice Display Signs of Stress Erythropoiesis When Fed an Iron-Deficient Diet

To further examine hepcidin regulation in mice lacking functional MT2, 10 week-old *mask* mice were placed on an iron-deficient diet for two months. This treatment further decreased the already low hematocrit and hemoglobin values. Spleen weights in *mask* mice fed the iron-deficient diet were significantly increased (Figure 4) and flow cytometry analysis demonstrated increased numbers of basophilic erythroblasts and polychromatic erythroblasts (Table 1). The observed increase in splenic erythroid precursors and spleen weight indicates that *mask* mice on iron deficient diet activate stress erythropoiesis in the spleen, but, due to the systemic lack of iron, erythropoiesis is unable to proceed beyond the erythroblast stage.

In correlation with the increased spleen size, *mask* mice, but not C57BL/6 mice, responded to the administration of iron-deficient diet by increasing splenic erythroferrone (*Fam132b*) mRNA content (Figure 4). The increase in *Fam132b* mRNA was mirrored by an increase in spleen ERFE protein (Figure 5A). In addition to ERFE, *mask* mice on iron-deficient diet markedly increased the expression of splenic transferrin receptor (TFRC) and transferrin receptor 2 (TFR2) proteins; the increase in these two transferrin receptors is probably related to the increase in splenic erythroblast populations (Table 1), since both receptors are expressed in erythroblasts [27,28]. Although *mask* mice suffer from sideropenic anemia and their erythropoiesis is clearly iron-deficient, the content of splenic TFRC protein was not significantly different between C57BL/6 mice and *mask* mice on a standard diet (Figure 5B), despite the well-established post-transcriptional regulation of TFRC protein by intracellular iron [27]. Liver non-heme iron content in *mask* mice was substantially decreased in comparison with C57BL/6 mice and feeding of iron-deficient diet resulted in further decrease of liver iron concentration (Figure 6). Administration of iron-deficient diet to *mask* mice also resulted in a decrease in splenic iron concentration; however, the difference did not reach statistical significance (Table 2). Taking into account the increased spleen size, the total amount of splenic non-heme iron actually increased in *mask* mice on iron-deficient diet compared to *mask* mice on control diet (114 ± 50 µg vs. 53 ± 15 µg in males and 86 ± 20 µg vs. 75 ± 27 µg in females). Overall, the results show that *mask* mice kept on an iron-deficient diet are to some extent still able to support erythropoiesis, possibly by mobilizing iron from liver iron stores. At the same time, the decrease in hemoglobin concentration results in activation of stress erythropoiesis. In agreement with the observation that the main organ participating in mouse stress erythropoiesis is the spleen [29], spleens from *mask* mice kept on an iron-deficient diet displayed marked enlargement caused by the expansion of erythroblast populations; this increase in the number of erythroblasts resulted in substantial increase in splenic TFRC and TFR2 proteins. 

### 2.6. Mask Mice Downregulate Hepcidin When Fed an Iron-Deficient Diet

Feeding an iron-deficient diet to *mask* mice decreased the liver non-heme iron concentration; in addition, it also decreased liver *Hamp* mRNA content (Figure 6). Intriguingly, the decrease in *Hamp* expression was not associated with a decrease in BMP/SMAD signaling, as *Id1* expression was not significantly attenuated (Figure 6). These results indicate that some pathways mediating *Hamp* downregulation remain active in *mask* mice. 

### 2.7. Feeding an Iron-Deficient Diet Decreases Full-Length HJV Protein in Mask Mice

The expression of hepcidin is mediated by several proteins at the hepatocyte plasma membrane; therefore, it was of interest to determine whether feeding an iron deficient diet to *mask* mice will influence the liver content of HJV or NEO1. Immunoblotting of proteins present in the plasma membrane-enriched fraction from the liver demonstrated a decrease of the full-length HJV band in iron-depleted *mask* mice and also displayed a highly reproducible alternate cleavage of HJV [13], resulting in the appearance of two new *mask*-specific cleaved bands (Figure 7A). Thus, the absence of MT2 proteolytic domain resulted in an alternative pattern of HJV protein cleavage and similar pattern could be seen in *mask* mice on an iron-deficient diet (Figure 7A). The content of NEO1 protein was not different between the treatment groups (Figure 7B). A possible explanation for the decreased HJV protein content and the alternate pattern of its cleavage in *mask* mice would be the processing of HJV by another protease present at the hepatocyte membrane. It has already been demonstrated in vitro that HJV is a substrate for furin, and it has been reported that furin expression increases in iron deficiency and hypoxia [30]. However, despite low hemoglobin (Figure 4) and low non-heme liver iron content (Figure 6), the expression of *Furin* mRNA was not induced in *mask* mice fed an iron-deficient diet (Figure 7C), arguing against the involvement of furin in this experimental setting although post-transcriptional regulation of furin on the protein level cannot be excluded.

## 3. Discussion

Systemic iron metabolism is controlled by hepcidin. The most important signaling pathway which regulates the expression of hepcidin in response to iron availability is the BMP-SMAD signaling pathway [8,31]. A crucial component of hepatocyte BMP signaling is the hemojuvelin protein; in humans, mutations in the *HJV* gene can result in juvenile hemochromatosis that is clinically undistinguishable from hemochromatosis caused by mutations of the hepcidin gene itself [32].

The identification of matriptase-2 as a new negative regulator of hepcidin expression has immediately raised questions about the physiological substrates for its proteolytic activity. In vitro studies have identified HJV as a substrate for MT2 [7]. It is therefore rather intriguing that both *mask* mice and *Tmprss6*-deficient mice display decreased, rather than increased, total liver HJV protein content (Figure 1A and reference [21]). This observation makes it unlikely that the physiological function of MT2 is regulating the total amount of HJV protein at the hepatocyte plasma membrane. In agreement with this conclusion, literature data suggest that hepcidin expression in vivo apparently does not closely mirror changes in HJV protein content. For example, it has been noted that *Hjv*+/- mice display similar liver *Hamp* mRNA content as *Hjv* +/+ mice [33]; in addition, it has been reported that overexpression of HJV does not increase *Hamp* expression [18,34]. Of special interest, it has been noted that only 15% of the HJV levels of wild type animals was able to correct the severely reduced *Hamp* expression in *Hjv*−/− mice [34]. Taken together, these results suggest that, in mice, *Hamp* expression is not directly proportional to liver HJV protein content.

On the other hand, in support for the in vivo processing of HJV by MT2, we did previously report cleavage of liver HJV, coupled with increased content of MT2, in rats with severe iron deficiency anemia [23]. In rats, the content of liver *Tmprss6* mRNA is higher than in mice, and rat MT2 protein has been reported to increase already after three days on iron-deficient diet [35]. It is therefore possible that rat MT2 plays a more prominent role in regulating the content of plasma membrane hemojuvelin. It is interesting to note that the reported cleavage of HJV occurs close to the N-terminus [23], which is reported to participate in BMP6 protein binding [36]. For studies of MT2-dependent hepcidin regulation, rats appear to be the animals of choice; unfortunately, no model of MT2-deficient rat has been reported so far.

Recently published results indicate that the interaction between MT2 and HJV is based on non-proteolytic binding [19]. The importance of non-proteolytic protein-protein interactions is indirectly supported by the observation that the amount of full-length MT2 protein is substantially decreased in *Hjv*−/− mice (Figure 2A). In addition, the absence of HJV resulted in decreased amount of neogenin protein (Figure 2B). Therefore, it appears plausible that the mutual interaction between MT2, HJV and neogenin at the plasma membrane stabilizes these proteins, and that the absence of one protein results in more rapid degradation of the others.

One of the intriguing aspects of iron metabolism regulation in *Tmprss6*-deficient mice is the fact that these mice fail to downregulate hepcidin in the presence of anemia. Anemia has been reported to decrease hepcidin expression as early as 2002 [11]; subsequent research demonstrated that hepcidin expression responds to changes in erythropoietic activity [12,37] rather than to anemia itself. In wild-type adult mice, administration of erythropoietin (EPO) represents the most potent stimulus for hepcidin downregulation; however, *Tmprss6*−/− mice are completely resistant to EPO treatment [10]. Very probably, the inability of EPO to decrease hepcidin expression in *Tmprss6*−/− mice and *mask* mice is related to its inability to activate functional erythropoiesis in these animals. Administration of EPO to *Tmprss6*−/− mice appropriately stimulates their extramedullary erythropoiesis, as evidenced by increased spleen size [10]; however, the systemic lack of iron prevents the maturation of erythroblasts to hemoglobinized reticulocytes. When the systemic iron deficiency in *mask* mice is corrected by placing them on carbonyl iron diet, their red blood cell parameters normalize and administration of EPO results in a further increase of both hemoglobin and hematocrit (Figure 3B). These results demonstrate that the pathways mediating the response to EPO administration are intact in *mask* mice, but are hampered by profound systemic iron deficiency. Intriguingly, although EPO administration to iron-pretreated *mask* mice resulted in the expected increase in red blood cell parameters, it did not decrease the high expression of hepcidin (Figure 3A). The lack of effect of EPO on *Hamp* expression in iron-overloaded *mask* mice can be probably explained by the fact that the downregulation of hepcidin synthesis by erythropoietic activity occurs only when iron levels are within the physiological range, as previously demonstrated by the inability of EPO to decrease *Hamp* expression in iron-pretreated animals [6,38]. Conceivably, in this experimental setting, the amount of circulating ERFE is insufficient to sequester the increased amount of BMP proteins induced by dietary iron administration.

Although it is evident that EPO-mediated suppression of hepcidin expression is not functional in *Tmprss6*−/− mice [6] and *mask* mice [13], there is only limited information about the regulation of hepcidin in these mice by other stimuli. We therefore investigated the expression of hepcidin in *mask* mice fed long term an iron-enriched and iron-deficient diets. Administration of iron-enriched diet increased *Bmp6* expression, *Id1* expression and *Hamp* expression, confirming that the regulation of hepcidin by BMP6 functions normally in *mask* mice, as recently reported for *Tmprss6*-deficient mice [26]. This result is not unexpected, as all the components of the BMP/SMAD signaling pathway are present in *mask* mouse liver.

Interestingly, administration of iron-deficient diet to *mask* mice decreased *Hamp* expression, demonstrating that some pathways of hepcidin downregulation are active even in mice lacking MT2 proteolytic activity (Figure 6). From physiological point of view, the observed decrease in *Hamp* expression could be interpreted as an emergency response which allows at least some degree of iron mobilization from hepatocytes in order to partially support the iron deficient erythropoiesis. Feeding of iron-deficient diet to *mask* mice also resulted in marked increase in spleen weight, indicating activation of stress erythropoiesis. Spleen non-heme iron content was not significantly changed by iron deficiency, in contrast to liver non-heme iron concentration, which was downregulated. Possibly, the retention of iron in the spleen could reflect the fact that *mask* mice have sideropenic anemia which is reportedly associated with shortened erythrocyte life span [39], resulting in accelerated uptake of iron by splenic macrophages. Due to high hepcidin, the iron would be retained within the macrophages, leading to a state with apparently the same iron levels but, in fact, the level of biologically available iron for heme synthesis being limited in *mask* mice.

Feeding an iron-deficient diet also resulted in marked increase in splenic ERFE, TFRC and TFR2 protein expression, implying an increased demand for iron by the expanded (Table 1) splenic erythroblast populations. Theoretically, the increased synthesis of ERFE could have contributed to the observed downregulation of *Hamp* expression. However, since ERFE is reported to act through the BMP/SMAD pathway [17], the treatment should also result in decreased expression of *Id1*, which was not the case. It therefore appears that iron-deficient diet decreases *Hamp* expression in *mask* mice by a SMAD-independent mechanism. One of such mechanisms could be the post-transcriptional regulation of proteins known to play a role in iron metabolism, such as hemojuvelin, neogenin, HFE, TFR2 or the bone morphogenetic protein receptors. In this respect, it should be noted that the mechanisms responsible for hepcidin downregulation by iron deficiency are only incompletely understood and probably include changes in intracellular iron content and extracellular iron content [40], as well as possible post-transcriptional effects on the expression of plasma membrane proteins [41]. The observation that iron deficient diet decreases the HJV protein content in *mask* mice (Figure 7) suggests that the HJV protein could be posttranscriptionally regulated independently of MT2. Overall, the so far incomplete understanding of the interaction of MT2 with other hepatocyte plasma membrane proteins highlights the necessity for further immunoblot studies, which would greatly benefit from the availability of new validated antibodies suitable for in vivo use.

In conclusion, the presented study supports the interaction between MT2 and HJV proteins; however, the decreased total liver HJV protein content in *mask*-mice argues against simple MT2-mediated HJV cleavage as a physiological mechanism by which MT2 regulates hepcidin expression. Experiments with long term iron-deficient diet show that *mask* mice do respond differently compared to wild type mice and induce stress erythropoiesis coupled with increased production of ERFE, TFRC and TFR2 in the spleen. Our data suggest that *mask* mice are able to produce immature erythroid precursors but cannot properly hemoglobinize them into mature erythrocytes. The observed decrease of liver HJV protein in *mask* mice fed an iron-deficient diet suggests possible processing of HJV by another protease, and underscores the necessity for further studies aimed at such protein-protein interactions.

## 4. Materials and Methods

### 4.1. Animals and Treatment

Animal experiments were approved by Czech Academy of Sciences Experimental Project no. 25/2015 and Czech Ministry of Education Protocol 11192/2020-2. *Mask* mice were obtained from Mutant Mouse Resource Research Centers, West Sacramento, CA, USA. Breeding pairs were kept on an iron-enriched diet (Harlan Teklad, Indianapolis, IN, USA, 2% carbonyl iron, TD.09521); the same diet fed for 2 months to adult mice was used for iron overload experiments. Weaned *mask* mice were kept on a standard laboratory diet (iron content approximately 200 ppm). Littermates with intact *Tmprss6* genes (C57BL/6 background) were used for comparison. EPO (NeoRecormon Roche, 2000 U) was administered once daily at 50 U/mouse for four days. For experiments with iron-deficient diet, Altromin C1038 (Lage, Germany, iron content approximately 5 ppm) was used. C57BL/6J and *mask* mice were placed on iron-deficient diet at 10 weeks of age; duration of feeding of the iron-deficient diet was two months. The purpose of the experiments with iron-deficient diet was to induce only moderate iron deficiency in *mask* mice, without dramatic worsening of their iron deficiency anemia, which could possibly endanger the animals. Therefore, 10 week old mice were selected rather than the more commonly used rapidly growing young mice [42], which have a higher demand for iron. C57BL/6J mice were obtained from commercial sources (Velaz SRO and Anlab SRO, Prague, Czech Republic).

### 4.2. Immunoblotting

Liver samples for immunoblotting were prepared either as whole liver homogenates, or as plasma membrane-enriched fractions. Whole homogenates were prepared by Ultra Turrax homogenization (6 mm, 3 × 10 s) in TRIS-buffered (pH 8) 150 mM NaCl containing 1% of Igepal Ca-630 detergent (Sigma Aldrich, Prague, Czech Republic). Plasma membrane-enriched fractions were obtained by centrifugation of liver homogenates prepared in 10 mM HEPES buffer without the addition of detergent at 3000 g followed by extensive washing, as detailed in [23]. Electrophoresis was run on 8% (MT2) or 10% (HJV) SDS- polyacrylamide gels under denaturing and reducing conditions, samples were heated for 10 min at 85 °C prior to loading. PVDF membranes were used for blotting in Invitrogen SureLock apparatus. For HJV and MT2 determination, R&D AF3634 goat polyclonal antibody and Abcam ab56182 rabbit polyclonal antibody were used respectively; selectivity of both antibodies was verified on samples from knockout mice. NEO1 was determined by R&D 1079 goat polyclonal antibody. For the determination of phosphorylated SMAD proteins, liver homogenates were prepared with the addition of 1% of Phosphatase Inhibitor Cocktail 3 (Sigma Aldrich), phosphorylated SMADs were detected by Abcam ab92698 antibody.

For the detection of ERFE and transferrin receptors in the spleen, a membrane fraction obtained by ultracentrifugation was used. Spleen samples (approximately 50 mg) were homogenized in 1 mL of 10 mM HEPES buffer containing 250 mM of sucrose and centrifuged for 15 min at 8000× *g*. The supernatant was then subjected to centrifugation at 105,000× *g* for 1 h; the pellet was washed once by recentrifugation in fresh buffer and resuspended in 80 µl of 2% SDS containing 25 mM of ammonium bicarbonate. This procedure allowed the detection of ERFE associated with the endoplasmic reticulum membranes. ERFE was detected by the discontinued Santa Cruz sc-246567 antibody; TFRC was detected using Abcam ab214039 antibody at 1:1000 dilution, TFR2 was detected by Alpha Diagnostics TFR21-A antibody at 1:500 dilution.

### 4.3. Gene Expression Analyses

RNA content was determined by real-time PCR on Biorad IQ8 cycler following extraction by Qiagen RNeasy Plus Mini Kit and reverse transcription by ThermoFisher RevertAid Kit. For all PCR analyses, *Actb* was used as a reference gene. Primer sequences are given in Appendix A.

For positive control immunoblot analyses, liver samples from wild-type mice overexpressing HJV from an adeno-associated virus under the control of a liver-specific promoter (AAV2/8-*Hjv-Flag*) were used; the samples showed the overexpressed HJV band. The overexpression of *Hjv* mRNA was confirmed by quantitative real-time PCR.

### 4.4. Flow Cytometry

Spleen tissue was disrupted by a loosely fitting glass homogenizer, single-cell suspension was obtained by repeated passage through a 25G needle and filtered through a 70 μm nylon cell filter. Cells were stained by fluorescently labeled antibodies for 20 min at 4 °C in the dark and subsequently analyzed with a FACSAria IIu cell sorter (BD Biosciences, Franklin Lakes, NJ, USA). Populations of erythrocyte precursors were identified based on the Ter119/CD44 pattern [43].

### 4.5. Hematological Parameters and Iron Content

Hemoglobin was determined on Mindray BV 5300 Vet hematologic analyzer. Hematocrit values were determined by centrifugation in microtubes. Liver non-heme iron content was determined according to [44]. Plasma iron content was determined by a commercial kit (Fe Liquid 200, Erba-Lachema, Brno, Czech Republic).

## Figures and Tables

**Figure 1 ijms-22-02650-f001:**
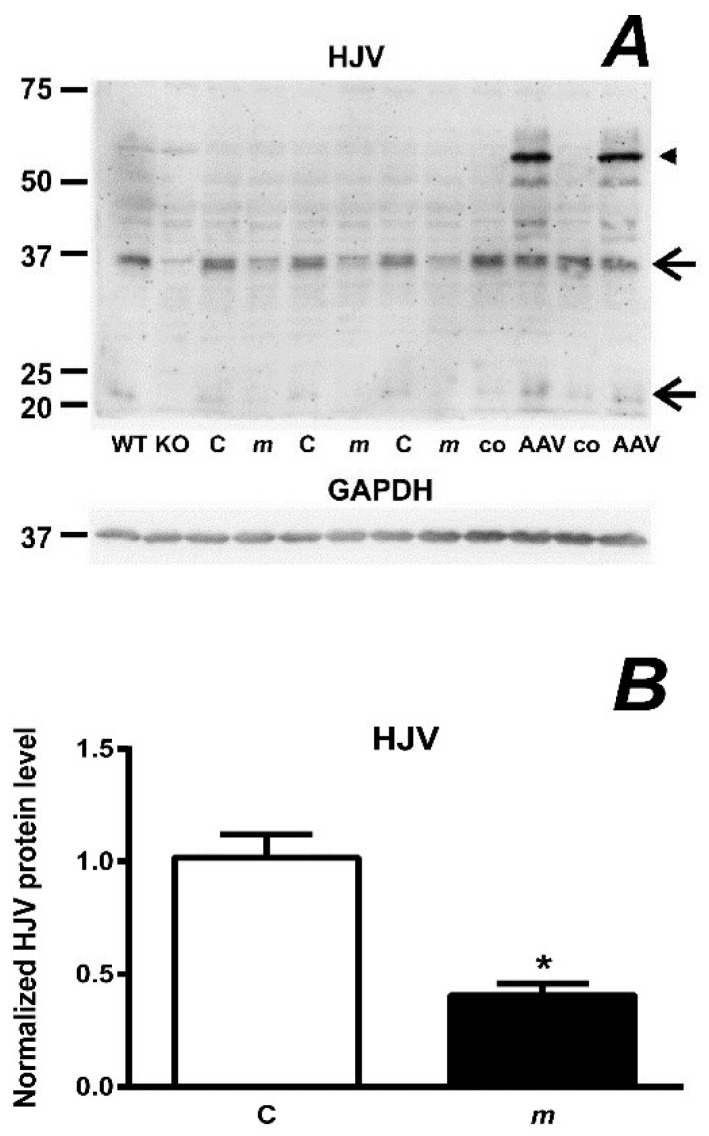
(**A**): Comparison of HJV protein content in detergent-containing homogenates from C57BL/6 mice (C) and *mask* mice (*m*). The specificity of the AF3634 anti-HJV antibody is demonstrated in the first two lanes by samples prepared in the same way from wild-type (WT) and *Hjv*−/− mice (KO). Samples prepared from mice injected with AAV2/8-*Hjv-Flag* vector (AAV) and control (co) mice are included on the right. Arrows denote the two endogenous HJV bands, arrowhead denotes the main HJV band in HJV-overexpressing mice. GAPDH is used as loading control. (**B**): Densitometric analysis of a 35 kDa form of HJV in control (C) and *mask* (*m*) mice. Asterisk (*) denotes statistical significance from the respective control group (*p* ˂ 0.05, *n* = 3).

**Figure 2 ijms-22-02650-f002:**
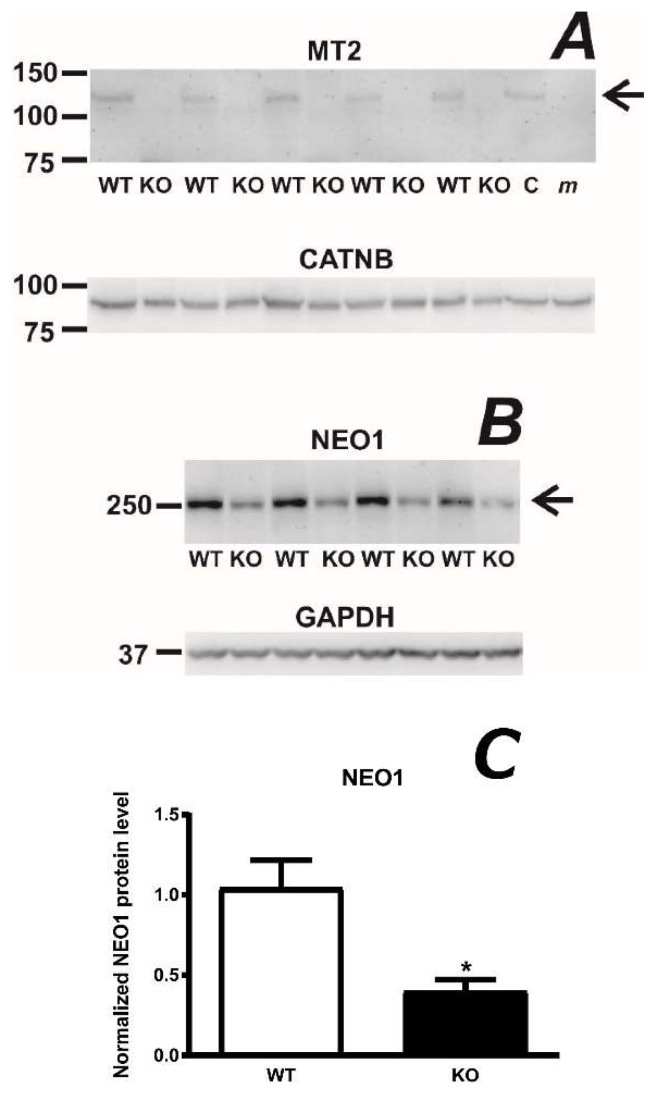
(**A**): Comparison of full-length MT2 protein content (arrow) in plasma membrane-enriched samples from *Hjv*+/+ (WT) and *Hjv*−/− (KO) mice. The specificity of the ab56182 anti-MT2 antibody is demonstrated by samples prepared in the same way from C57BL/6 (C) and *mask* mice (*m*). β-catenin (CATNB) is used as loading control. (**B**): Comparison of neogenin (NEO1) protein (arrow) content in detergent-containing liver lysate from *Hjv*+/+ (WT) and *Hjv*−/− (KO) mice. GAPDH is used as loading control. (**C**) Densitometric analysis of NEO1 of *Hjv*+/+ (WT) and *Hjv*−/− (KO) mice. Asterisk (*) denotes statistical significance from the respective control group (*p* ˂ 0.05, *n* = 4).

**Figure 3 ijms-22-02650-f003:**
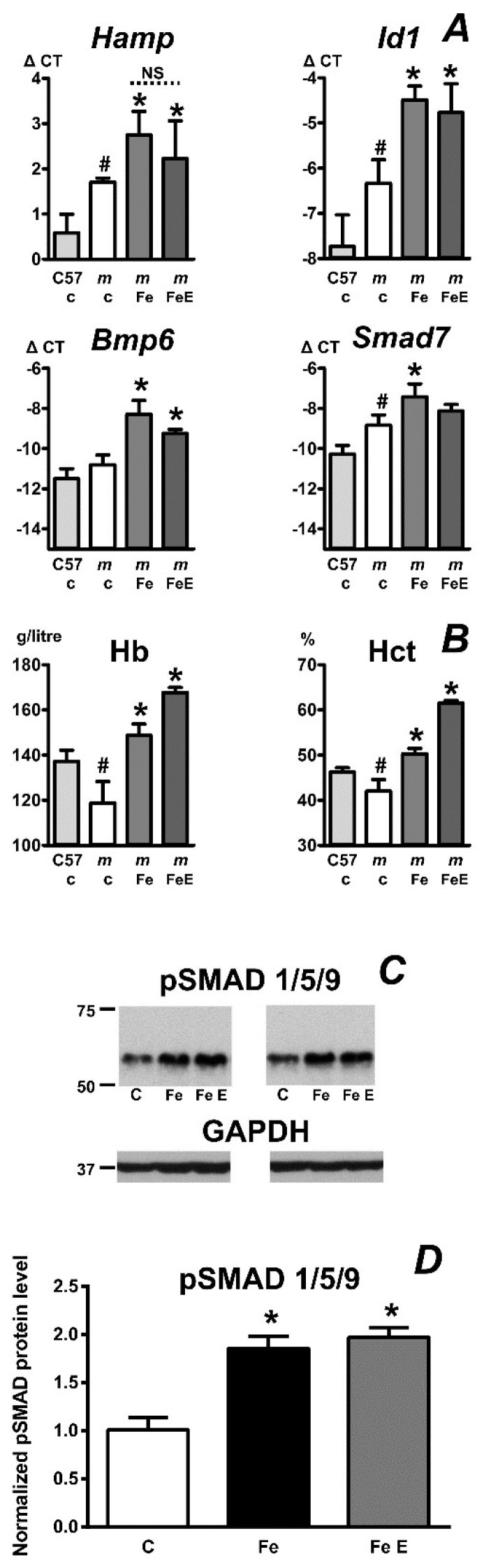
(**A**): Relative expression of *Hamp*, *Id1*, *Bmp6* and *Smad7* mRNA in *mask* mice fed a control diet (c), iron-enriched diet (Fe) or iron-enriched diet in combination with erythropoietin administration (FeE). Iron diet (2% of carbonyl iron) was administered for two months, erythropoietin was injected once daily at 50 U/mouse for the last four days of the experiment. ΔCT values are calculated relative to *Actb* mRNA content. Asterisks (*) denote statistically significant difference (*p* ˂ 0.05, *n* = 3) between *mask* mice on control diet and iron-enriched diet and hashtags (#) denote statistically significant difference (*p* ˂ 0.05, *n* = 3) between control C57BL/6 and *mask* mice. NS denotes a non-significant difference. Data from C57BL/6 mice on control diet are included for comparison. (**B**): Effect of the treatment on hemoglobin content (Hb) and hematocrit (Hct). (**C**): Effect of the treatment on phosphorylated SMAD 1/5/9 protein content in *mask* mouse liver homogenates. C, control; Fe, iron-enriched diet for two months; Fe E, iron-enriched diet in combination with EPO-treatment. (**D**) Densitometric analysis of pSMAD 1/5/9 in *mask* mice from panel C. Asterisks denote statistical significance from the respective control group (*p* ˂ 0.05, *n* = 2).

**Figure 4 ijms-22-02650-f004:**
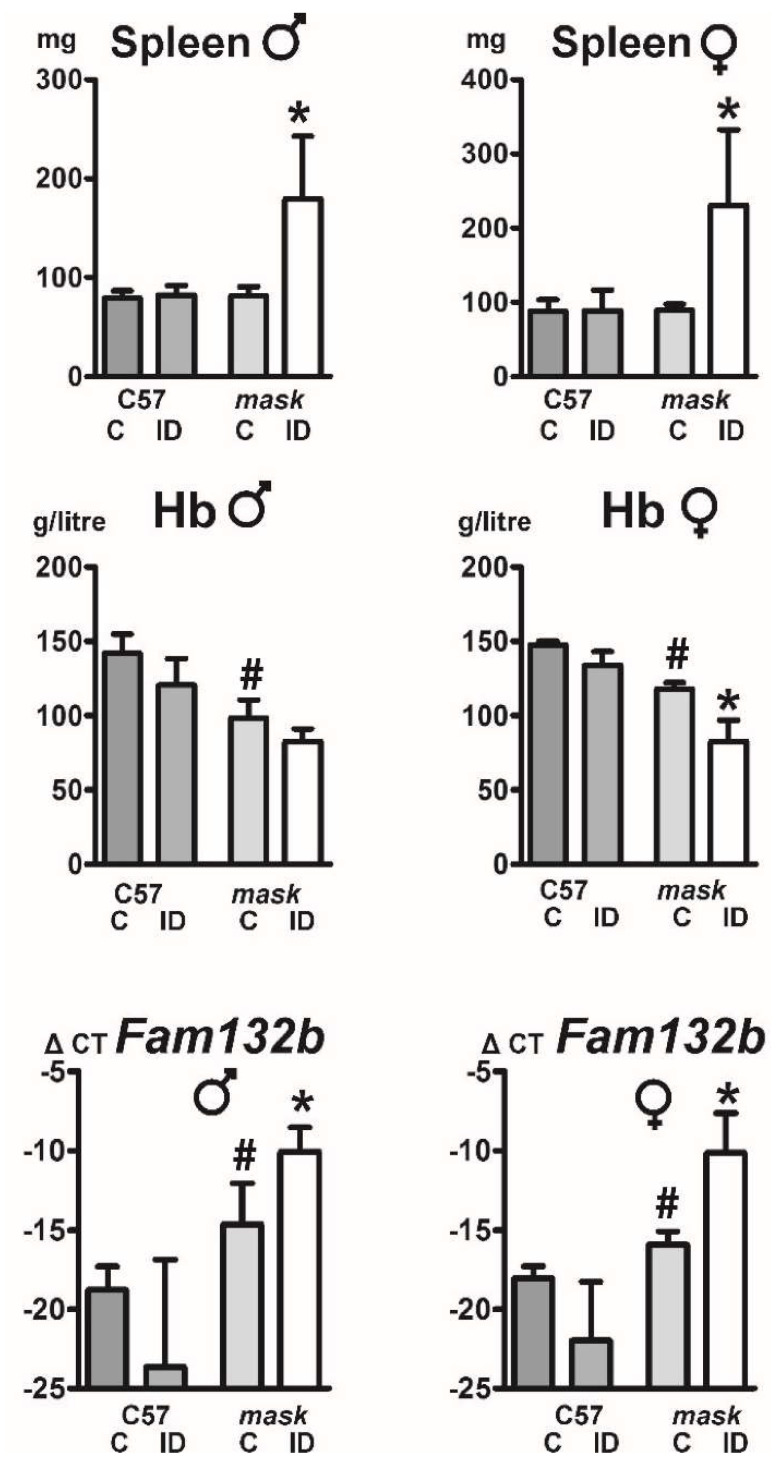
Effect of iron-deficient diet on spleen size, hemoglobin (Hb) concentration and *Fam132b* expression in male and female C57BL/6 mice and *mask* mice. Control (C) or iron-deficient (ID) diet was administered to 10 week old mice for 2 months. Asterisks (*) denote statistical significance (*p* ˂ 0.05, *n* = 3) from the respective control diet group and hashtags (#) denote statistically significant difference (*p* ˂ 0.05, *n* = 3) between control C57BL/6 and *mask* mice. ΔCT values are calculated relative to *Actb* mRNA content.

**Figure 5 ijms-22-02650-f005:**
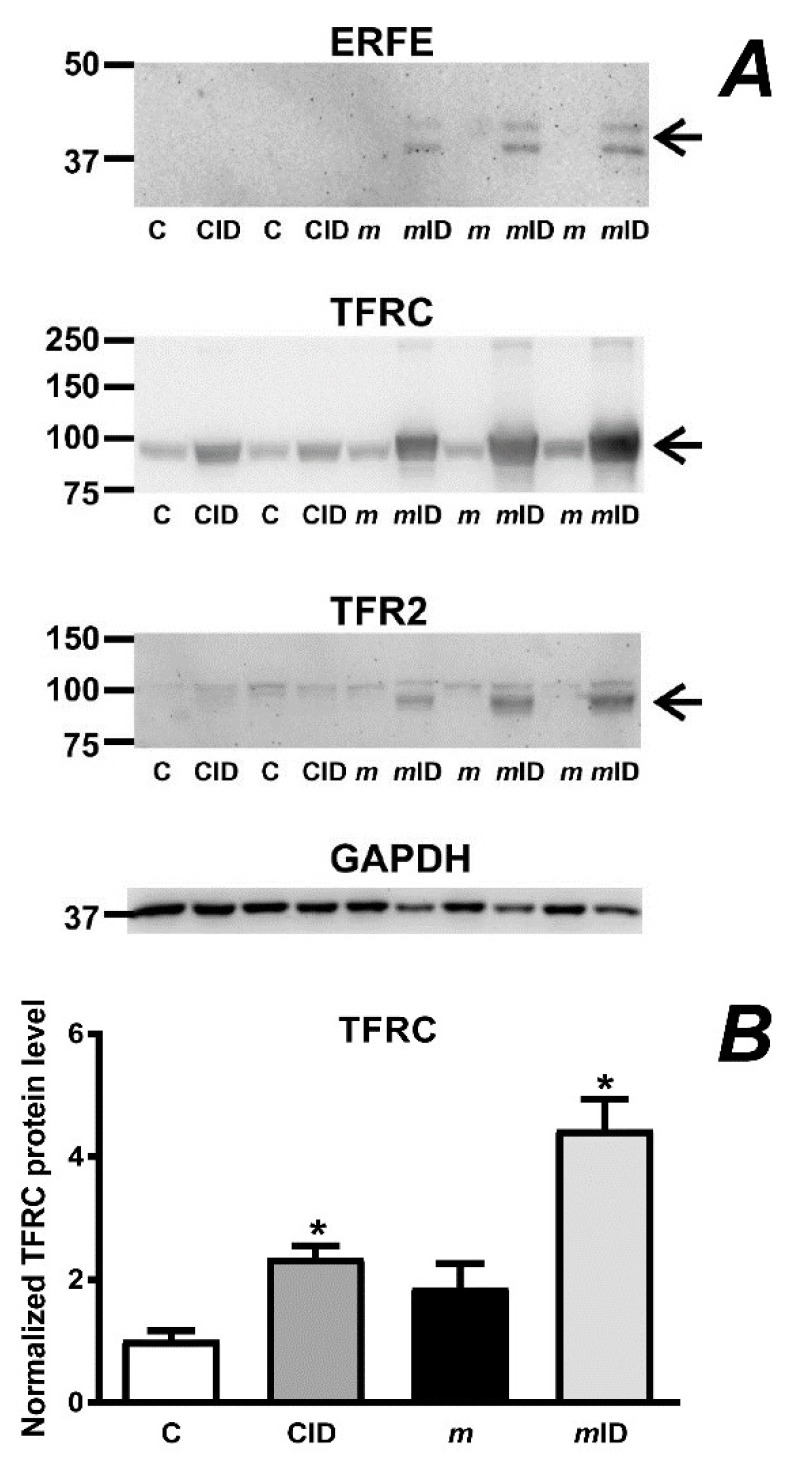
(**A**) ERFE, TFRC and TFR2 protein content in spleen membranes from C57BL/6 mice (C), C57BL/6 mice placed on an iron-deficient diet (CID), *mask* mice (*m*) and *mask* mice placed on an iron-deficient diet (*m*ID). Presented results are from female mice; results from male mice are given in Appendix A. Arrows denote the relevant bands. TFRC and TFR2 are detected as single bands; ERFE is detected as a double band. GAPDH is used as loading control. (**B**) Densitometric analysis of TFRC. Asterisks (*) denote statistical significance from the respective control group (*p* ˂ 0.05, *n* = 2–3). Densitometry of ERFE and TFR2 was not determined as there was no detectable expression in controls.

**Figure 6 ijms-22-02650-f006:**
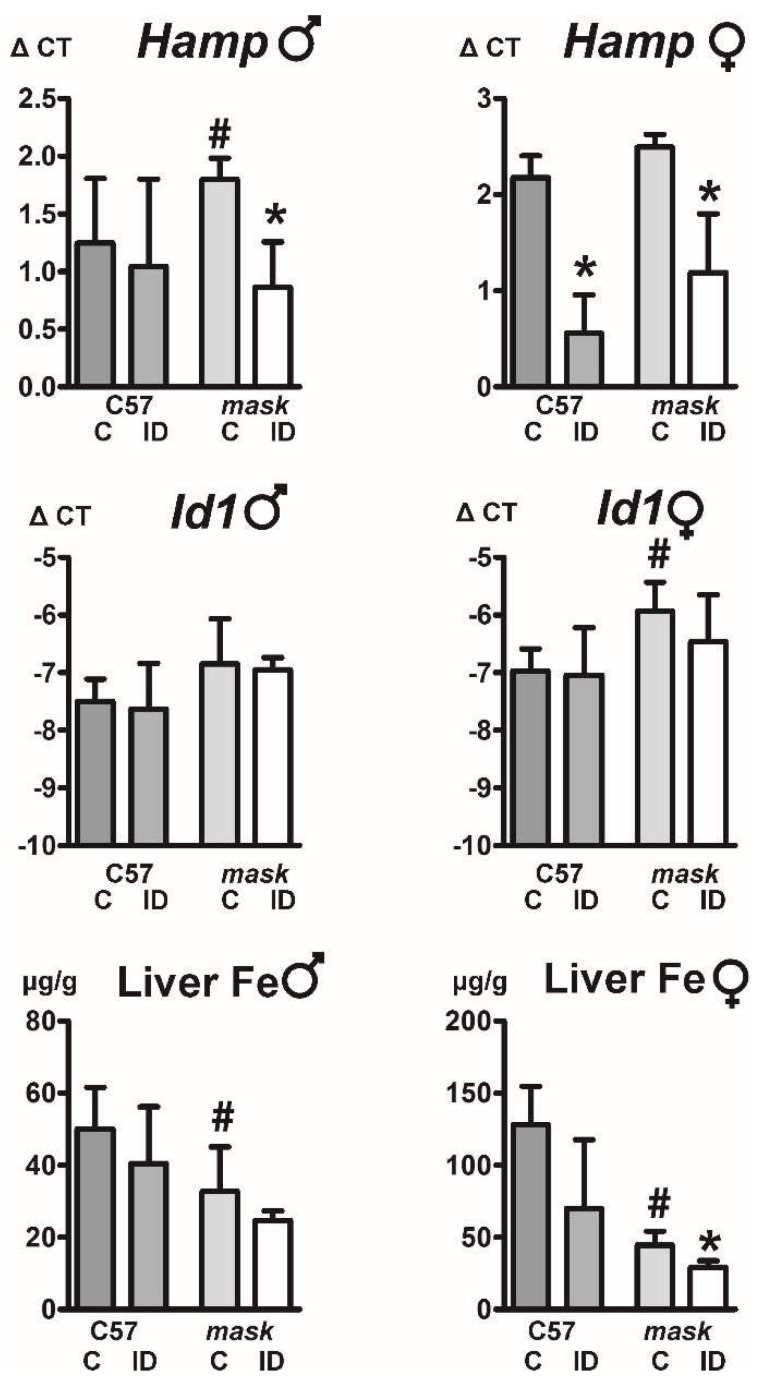
Effect of iron-deficient diet on liver *Hamp* expression, *Id1* expression and liver non-heme iron concentration. Control (C) or iron-deficient (ID) diet was administered to 10 week old C57BL/6 mice (C57) and *mask* mice for two months. Asterisks (*) denote statistical significance from the respective control diet group (*p* ˂ 0.05, *n* = 3) and hashtags (#) denote statistically significant difference (*p* ˂ 0.05, *n* = 3) between control C57BL/6 and *mask* mice. Δ CT values are calculated relative to *Actb* mRNA content, liver iron concentration is expressed as μg/g wet weight.

**Figure 7 ijms-22-02650-f007:**
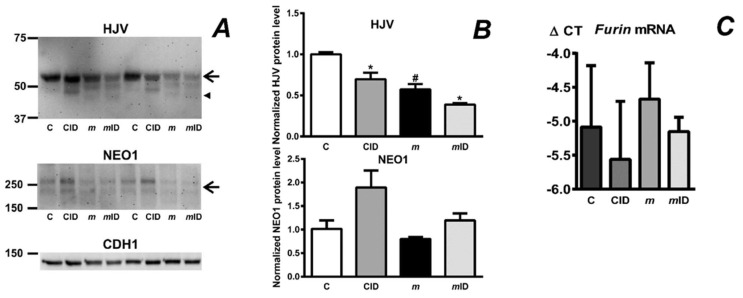
(**A**) Expression of HJV and NEO1 proteins in C57BL/6 mice on control diet (C), C57BL/6 mice on iron-deficient diet (CID), *mask* mice (*m*) and *mask* mice on iron-deficient diet (*m*ID). Proteins were determined in the 3000 g plasma membrane-enriched fraction. Presented results are from female mice; results from male mice are given in Appendix A. Arrows indicate the full-length HJV protein band and the double neogenin protein band, arrowhead indicates the cleaved HJV band seen in *mask* mice. (**B**) Densitometric analysis of full-length HJV and NEO1 protein content. Asterisks (*) denote statistical significance from the respective control diet group (*p* ˂ 0.05, *n* = 3–5) and hashtags (#) denote statistically significant difference (*p* ˂ 0.05, *n* = 3–5) between control C57BL/6 and *mask* mice. (**C**) Effect of iron deficient diet on liver *Furin* mRNA expression. Δ CT values are calculated relative to *Actb* mRNA content, *n* = 3.

**Table 1 ijms-22-02650-t001:** Populations of spleen erythroid precursors in *mask* mice fed an iron-deficient diet.

Group	ProE %	Baso %	Poly %	Ortho + Reti %	RBC %
C57BL/6	0.16 ± 0.06	3.76 ± 1.96	9.21 ± 0.01	17.45 ± 6.93	69.42 ± 8.95
*mask*	0.17 ± 0.02	2.84 ± 0.91	4.91 ± 1.25	19.58 ± 2.37	72.51 ± 4.56
*mask* ID	0.37 ± 0.12	11.25 ± 1.68	15.98 ± 0.95	30.40 ± 0.30	42.01 ± 0.55

Male *mask* mice were fed control or iron-deficient diet (ID) for two months; erythroid cells precursors were identified by flow cytometry based on the Ter119/CD44 pattern as described in Materials and Methods. Table shows relative percentages of proerythroblasts (ProE), basophilic erythroblasts (Baso), polychromatophilic erythroblasts (Poly), orthochromatic erythroblasts plus reticulocytes (Ortho + Reti) and mature red blood cells (RBC). Data represent mean ± standard deviation from two experiments.

**Table 2 ijms-22-02650-t002:** Spleen and plasma iron concentration in *mask* mice fed an iron-deficient diet.

Group	Spleen Fe, Males (µg/g)	Spleen Fe, Females (µg/g)	Plasma Fe, Males (µmol/L)	Plasma Fe, Females (µmol/L)
C57BL/6	706 ± 192	1356 ± 54	22.3 ± 3.1	25.7 ± 3.5
C57BL/6 ID	243 ± 66 *	478 ± 495	26.3 ± 1.9	21.3 ± 3.5
*mask*	643 ± 174	814 ± 300	5.2 ± 3.0 #	8.7 ± 2.1 #
*mask* ID	514 ± 166	480 ± 214	4.0 ± 1.0	3.7 ± 0.6 *

Male and female 10 week old C57BL/6 and *mask* mice were fed control or iron-deficient diet (ID) for two months. Iron concentration was determined as described in Materials and Methods, spleen iron concentration is expressed per wet weight. Data represent mean ± standard deviation. Asterisks (*) denote significant difference between control mice and mice kept on an iron-deficient diet, hashtags (#) denote statistically significant difference between control C57BL/6 and *mask* mice (*p* < 0.05, *n* = 3).

## Data Availability

Data is contained within the article or Appendix A.

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
