# Peer review of "Matriptase-2 and Hemojuvelin in Hepcidin Regulation: In Vivo Immunoblot Studies in Mask Mice"

_ijms, 2021, doi:10.3390/ijms22052650_

Round 1

Reviewer 1 Report

Authors of the article submitted for the review “Matriptase-2 and hemojuvelin in hepcidin regulation: In vivo immunoblot studies in mask mice" investigate the interaction of matriptase-2 with hemojuvelin protein in vivo. To characterize the role of matriptase-2, they investigated iron metabolism in mask mice fed experimental iron reach and iron-deficient diets. I totally agree with Author’s opinion that “The purpose of the current report is to summarize the in vivo results obtained on the interaction between MT2 and HJV, and to present new data on hepcidin regulation by iron overload and deficiency in Tmprss6-mutated mice”. Therefore, in my opinion the results of the presented research seems to be more descriptive and repetitive than novel. Many of the research results have been confirmed elsewhere, so the manuscript reads like a review article rather than an original research paper. Highly speculative publication. The most interesting result of the manuscript is the demonstration that the pathways mediating the response to EPO administration are intact in mask mice, but are hindered by deep iron deficiency. Second, EPO administration to iron-pretreated mask mice increased red blood cell indices, but did not decrease the high expression of hepcidin.

Minor requests:

  1. Taking above the Title should point methodological aspects of the paper. But decision is up to Authors.
  2. As it was shown by Wahedi et al, 2017, MT2 may cleave not only HJV, but other components of hepcidin expression pathway. Why neo and pSMAD have been analyzed selectively in the separate feeding experiments?
  3. Assuming that proteins interact with each other, why have no lipid rafts been isolated where there is physical contact between the proteins on the membranes.
  4. Furthermore, to support the hypothesis that HJV, MT2, and NEO1 form a complex on the hepatocyte plasma membrane, why was the immunofluorescent localization of these complexes not performed? This would improve the quality of the research in the manuscript.
  5. In the chapter 2.4. Authors claim that in mice fed with a diet containing 2% of CI the hepatic level of iron increased. What was the control hepatic non-heme iron content in mask mice? In the Fig.6 it was shown approx. 50 µg/g in females mask mice, while in chapter 2.4. 100 µg/g.
  6. What was the level of non-heme iron in the spleen? Similar non-hem iron values could explain the lack of changes in TFRC protein level between C57BL/6 mice and mask mice on a standard diet. In this case, IRP1 activity could be the same (due to the same iron content) and in the same way regulate IRE-containing proteins. In addition, IRE/IRP activity analysis (EMSA) would be helpful to resolve this issue.
  7. Iron-deficient diet induce splenomegaly in mask mice but not in wt mice. This supports the idea of the erythropoietic activity in the spleen in this mice. However, to coordinate iron supply with Hb synthesis, erythropoietin stimulated erythroid precursors in both BM and the spleen release erythroferrone, a soluble protein that suppresses the expression of hepcidin. Other words, why the spleen was the only tissue, while ERFE serum content is a result of both spleen and BM expression? Serum ELISA ERFE analysis would be preferred. Likewise for serum levels of hepcidin.
  8. Why serum iron level results haven’t been reported?

In conclusion, the presented study supports the interaction between MT2 and HJV proteins, but the key question about the mechanism of such interaction remains unanswered. Unfortunately, this manuscript does not bring the Authors closer to this answer. I hope that my suggestions will be helpful in getting the final version of the publication.

Author Response

Authors of the article submitted for the review “Matriptase-2 and hemojuvelin in hepcidin regulation: In vivo immunoblot studies in mask mice" investigate the interaction of matriptase-2 with hemojuvelin protein in vivo. To characterize the role of matriptase-2, they investigated iron metabolism in mask mice fed experimental iron reach and iron-deficient diets. I totally agree with Author’s opinion that “The purpose of the current report is to summarize the in vivo results obtained on the interaction between MT2 and HJV, and to present new data on hepcidin regulation by iron overload and deficiency in Tmprss6-mutated mice”. Therefore, in my opinion the results of the presented research seems to be more descriptive and repetitive than novel. Many of the research results have been confirmed elsewhere, so the manuscript reads like a review article rather than an original research paper. Highly speculative publication. The most interesting result of the manuscript is the demonstration that the pathways mediating the response to EPO administration are intact in mask mice, but are hindered by deep iron deficiency. Second, EPO administration to iron-pretreated mask mice increased red blood cell indices, but did not decrease the high expression of hepcidin.

Minor requests:

1. Taking above the Title should point methodological aspects of the paper. But decision is up to Authors.

We believe that the Title reflects the methodology as it states that those are mainly immunoblot studies.

2. As it was shown by Wahedi et al, 2017, MT2 may cleave not only HJV, but other components of hepcidin expression pathway. Why neo and pSMAD have been analyzed selectively in the separate feeding experiments?

 Neogenin 1 has been analyzed as a possible constituent of the HJV/TMPRSS6 complex and pSMAD was probed to reflect the activity of the BMP/SMAD pathway. If I interpret the request correctly, the reviewer would like to see both proteins at the same time in Figure 7. We performed the blot for pSMAD1/5/9 on the iron deficient diet, but there was no significant change, corresponding with the mRNA data from Figure 6 and thus we are not showing the blot. We have modified the result section on lines 284-288 to “Feeding of iron-deficient diet to mask mice decreased the liver iron concentration; in addition, it also decreased liver Hamp mRNA content (Figure 6). Intriguingly, the decrease in Hamp expression was not associated by a decrease in BMP/SMAD signaling, as Id1 expression was not significantly attenuated (Figure 6) and pSMAD1/5/9 protein level was not changed as well (data not shown).”

3. Assuming that proteins interact with each other, why have no lipid rafts been isolated where there is physical contact between the proteins on the membranes.

We do not routinely isolate lipid rafts in the laboratory and we would like to thank the reviewer for that point; however, due to time requirements, we could not perform the experiment. Furthermore, we believe that the isolation of the plasma membranes that was used actually also enriches the plasma membrane rafts, so to some extent the results we present on plasma membrane could parallel the results obtained by isolation of the lipid rafts specifically, but, of course, that needs further experimental confirmation.

4. Furthermore, to support the hypothesis that HJV, MT2, and NEO1 form a complex on the hepatocyte plasma membrane, why was the immunofluorescent localization of these complexes not performed? This would improve the quality of the research in the manuscript.

We agree with the reviewer that such piece of information would help the manuscript; however, for this we would have to have a fixed piece of tissue and specific antibodies that work with immunofluorescence procedure. Unfortunately, it is quite likely that the antibody against TMPRSS6 that works on murine samples and gives some unspecific bands on western blot is quite unlikely to give specific signal that with immunofluorescence staining. Again, given the time requirements of the revision did not give us the option to perform the requested analyses.

5. In the chapter 2.4. Authors claim that in mice fed with a diet containing 2% of CI the hepatic level of iron increased. What was the control hepatic non-heme iron content in mask mice? In the Fig.6 it was shown approx. 50 µg/g in females mask mice, while in chapter 2.4. 100 µg/g.

We thank the reviewer for pointing this out, it was the mistake as the correct values are app. 50 µg/g, The sentence has been corrected and now stands as” Feeding of a diet containing excessive amount of iron (2 % as carbonyl iron) to female mask mice for two months dramatically increased liver non-heme iron (from approximately 50 µg/g to over 1500 µg/g wet weight).”

6. What was the level of non-heme iron in the spleen? Similar non-hem iron values could explain the lack of changes in TFRC protein level between C57BL/6 mice and mask mice on a standard diet. In this case, IRP1 activity could be the same (due to the same iron content) and in the same way regulate IRE-containing proteins. In addition, IRE/IRP activity analysis (EMSA) would be helpful to resolve this issue.

The values of non-heme iron in the spleen are not significantly different between the control and mask mice irrespective of sex. This is in contrast to liver iron, where both males and females show a decrease in liver non-heme iron level. Possibly, it could reflect the fact that mask mice have hypochromic microcytic anemia and their erythrocytes might have a shortened life span and could be taken up by splenic macrophages. Due to higher hepcidin, the iron would be retained in them, leading to a state with apparently the same iron levels but, in fact, the level of biologically available iron for heme synthesis is limited in mask mice. This phenotype is probably more exacerbated under iron deficiency where these mice exhibit stress erythropoiesis. Unfortunately, due to the revision period of 5 days we cannot perform the suggested EMSA assay. Yet, we have modified the discussion and included the sentence on lines 405–413 stating ” Interestingly, the spleen non-heme iron content was not changed by iron deficiency, in contrast to liver non-heme iron level, which was downregulated. Possibly, it could reflect the fact that mask mice have hypochromic microcytic anemia and their erythrocytes might have a shortened life span, resulting in the uptake by splenic macrophages. Due to higher hepcidin, the iron would be retained within the macrophages, leading to a state with apparently the same iron levels but, in fact, the level of biologically available iron for heme synthesis being limited in mask mice. This phenotype is probably more exacerbated under iron deficiency where mask mice exhibit stress erythropoiesis, but it needs experimental validation.”

7. Iron-deficient diet induce splenomegaly in mask mice but not in wt mice. This supports the idea of the erythropoietic activity in the spleen in this mice. However, to coordinate iron supply with Hb synthesis, erythropoietin stimulated erythroid precursors in both BM and the spleen release erythroferrone, a soluble protein that suppresses the expression of hepcidin. Other words, why the spleen was the only tissue, while ERFE serum content is a result of both spleen and BM expression? Serum ELISA ERFE analysis would be preferred. Likewise for serum levels of hepcidin.

Unfortunately, we were not able to collect the bone marrow in this experimental setup due to technical reasons and due to the time requirements to submit revised version within 5 days we cannot repeat the experiment and perform the ERFE ELISA, which would be helpful. Yet, based on our previous work we can speculate that the response of the bone marrow will be similar to that of spleen as we have seen a coordinated production of ERFE by both compartments.

8. Why serum iron level results haven’t been reported?

 Unfortunately, the serum iron has not been measured due to technical reasons and we were not able to measure them within the revision period.

In conclusion, the presented study supports the interaction between MT2 and HJV proteins, but the key question about the mechanism of such interaction remains unanswered. Unfortunately, this manuscript does not bring the Authors closer to this answer. I hope that my suggestions will be helpful in getting the final version of the publication.

Reviewer 2 Report

Data presented in this work contribute to revealing that MT2, HMJ, and neogenin protein levels are mutually regulated and that a redundant regulation of hemojuvelin stability exists apart of that exerted by MT2 protease activity. This alternative pathway possibly mitigates the effect of MT2 deficiency, but it’s not sufficient to normalize hepcidin synthesis.

Besides, authors report data indicating that mask mice are still able, upon iron administration, to respond to EPO in terms of erythropoiesis increase. In contrast, an iron deficient diet exacerbates their phenotype, inducing significant stress erythropoiesis.

Specific points

Figure 1 of this document and figure 6 of ref 13 by the same authors seem redundant. Authors should better clarify what novel message is reported in this manuscript in Figure 1. Otherwise, they should remove Figure 1 from the current manuscript. Reduction of HMJ protein content in mask mice will be still showed in figure 7.

I appreciate that authors always show WB replicates of their results; however, in some cases, such as in figure 2A and figure 7, showing a densitometry summary with statistical analysis might help to understand the results.

It seems to me that mask mice fed a standard diet show a milder phenotype than that expected. Please, specify the amount of iron in your standard diet, and comment on any eventual difference with the composition of other standard diets used in literature. To better underline the effects of dietary iron modulation and EPO administration in mask mice, please indicate any statistical difference occurring between mask and wt mice fed a standard diet.

Author Response

Comments and Suggestions for Authors

Data presented in this work contribute to revealing that MT2, HMJ, and neogenin protein levels are mutually regulated and that a redundant regulation of hemojuvelin stability exists apart of that exerted by MT2 protease activity. This alternative pathway possibly mitigates the effect of MT2 deficiency, but it’s not sufficient to normalize hepcidin synthesis.

Besides, authors report data indicating that mask mice are still able, upon iron administration, to respond to EPO in terms of erythropoiesis increase. In contrast, an iron deficient diet exacerbates their phenotype, inducing significant stress erythropoiesis.

Specific points

Figure 1 of this document and figure 6 of ref 13 by the same authors seem redundant. Authors should better clarify what novel message is reported in this manuscript in Figure 1. Otherwise, they should remove Figure 1 from the current manuscript. Reduction of HMJ protein content in mask mice will be still showed in figure 7.

Thanks to the reviewer for pointing this out, we have deleted panel B of Figure1 so as it is not duplicated in Figure7. Panel A shows the verification with Hjv KO mice as well as Hjv overexpressing mice and thus was left as is. We have modified the manuscript to reflect those changes (deleted former 2.2 section of the manuscript). The description of alternative cleavage of hemojuvelin is now at lines 294-300, reading as follows ”Immunoblotting of proteins present in the plasma membrane-enriched fraction from the liver demonstrated a decrease of the full-length HJV band in iron-depleted mask mice and also displayed a highly reproducible alternate cleavage of HJV [13], resulting in the appearance of two new mask-specific cleaved bands (Figure 7A). Thus, the absence of MT2 proteolytic domain resulted in an alternative pattern of HJV protein cleavage and similar pattern could be seen in mask mice on an iron-deficient diet (Figure 7A).”

I appreciate that authors always show WB replicates of their results; however, in some cases, such as in figure 2A and figure 7, showing a densitometry summary with statistical analysis might help to understand the results.

The densitometry was performed using the AzureSpot 2.0 software and the graphical evaluation together with statistical evaluation is presented in Figure1, Figure2, Figure3, Figure5 and Figure7. In other cases, where there is no band in the control sample, it is not possible to make any assumptions as it is simply NO/YES situation and densitometry was thus not performed. Based on the densitometry results, we also modified the part of the manuscript concerning spleen TFRC content so it has been replaced by this statement on lines 227-231 “Although erythropoiesis in mask mice is clearly iron-deficient, the content of splenic TFRC protein was not significantly different in C57BL/6 mice and mask mice on a standard diet. The observed trend towards a higher TFRC level in mask mice is in line withpublished results, since the synthesis of TFRC protein is expected to increase in states of iron deficiency [27]”

It seems to me that mask mice fed a standard diet show a milder phenotype than that expected. Please, specify the amount of iron in your standard diet, and comment on any eventual difference with the composition of other standard diets used in literature. To better underline the effects of dietary iron modulation and EPO administration in mask mice, please indicate any statistical difference occurring between mask and wt mice fed a standard diet.

Statistical difference between the control and mask mice has been added to all relevant graphs and is marked with # sign, where it was statistically significant at a p value lower than 0.05. The standard diet used in this study is the altromin formula 1310 that contains 191 p.p.m. iron, so we believe it is in agreement with the published literature. Interestingly, we observed that mask males do have a stronger phenotype than female mice especially as they age. Mask males retain their alopecia throughout their lives while female mask mice seems to grow some hair as they age. It is possible that female mice are for some reason better at iron uptake from the diet and eventually achieve a milder phenotype than the males, yet the phenotype of hypochromic microcytic anemia is maintained.

Submission Date

31 December 2020

Date of this review

11 Jan 2021 01:39:04

Round 2

Reviewer 1 Report

Dear Editors and Authors. Unfortunately, I cannot say that the publication, after taking into account my comments, improved significantly in quality. The authors cannot explain the lack of opportunities to enrich the publication by the lack of time for additional analyzes. In this case, Editors should be asked for additional time to perform at least basic analyzes. I leave the final decision to the Editors.

Author Response

Response to Reviewer 1:

We thank the Reviewer for her/his comments. For the resubmission, we were able to add some of the requested data, which were unavailable for the first revision, since one of the authors (by ill luck the data curator) was at that time hospitalized. In addition, we made changes to the manuscript as a reaction to the specific points raised by the Reviewer.

One of the main concerns mentioned by the Reviewer is the repetitive nature of the results, which have already been confirmed elsewhere. We would like to stress the fact that every Figure contains some new data. It is true that Figure 1 repeats one of our main findings, namely the decreased content of HJV protein in mask mice. We have indeed published this (in our point of view very important) result previously. However, the decrease of HJV has recently been directly challenged (reference 19) by another group. It was for this reason that we re-published the results in Figure 1. In the resubmission, we have added new text to Section 2.1, which mentions and possibly explains this discrepancy.

Another point raised by the Reviewer is the highly speculative character of the manuscript. For the resubmission, we tried to avoid unfounded speculations and have added new references to support our statements. The main unconfirmed speculation of the manuscript is that the absence of HJV leads to increased degradation of MT2 and vice versa; however, the presented experimental data apparently support this conclusion. The theory that MT2, HJV and NEO1 form a complex at the hepatocyte plasma membrane has been published previously and our results only support it. We also state that the increased expression of splenic TFRC and TFR2 is probably related to the expansion of erythroblast populations. This statement is partly based on literature data, but is also strongly supported by our yet unpublished results on splenic TFRC and TFR2 expression in erythropoietin-treated mice.

Minor points:

  1. As to the Title, we believe that the current Title reflects the main methodology used, i.e. immunoblotting. By explicitly mentioning the method in the Title, we would very much like to emphasize the need for correct immunoblotting procedures in iron metabolism experiments.  
  2. The Wahedi 2017 paper mentions a number of possible MT2 targets: ALK2, ALK3, ActRIIA, Bmpr2, Hfe, HJV and TFR2. To our knowledge, apart from HJV and TFR2, validated antibodies against these targets which would be suitable for in vivo immunoblot use do not exist. For this reason, we selectively analyzed only HJV and NEO1 in the feeding experiments, as these two proteins are reported to form a complex with MT2.
  3. As pointed out in our previous response, we do not routinely isolate lipid rafts. Instead, we use the 3000 g plasma membrane-enriched fraction, which provides a relatively high protein yield (about 1 mg of protein/gram liver).
  4. As pointed out in our previous response, we do not regard the MT2 antibody sufficiently selective for immunofluorescence studies.
  5. Exact values are now included in Section 2.3.
  6. Splenic non heme iron content is now included in newly added Table 2. As pointed out previously, we do not routinely perform the IRE/IRP activity analysis.
  7. In experiments with low-iron diet, we interpreted the marked increase in spleen size as a marker of activated stress erythropoiesis. Since stress erythropoiesis in mice occurs mainly in the spleen, we concentrated on the spleen. To explain this approach, a new reference pointing to the role of the spleen in stress erythropoiesis was added to the manuscript (reference 28). Of course, we admit that the use of the spleen is also more practical, as it provides sufficient amount of material for analysis. Given the limited space of murine bone marrow and the marked increase in spleen size, it could probably be speculated that most of the new ERFE-producing erythroblasts originate in the spleen, although increased expression of Erfe in the bone marrow certainly takes place as well. As to the use of ERFE immunoblot rather than ERFE ELISA, we rather selfishly preferred the immunoblot, since, to our knowledge, our laboratory is the only one which has so far visualised ERFE in vivo on immunoblots, and the manuscript concentrates on immunoblots. From practical point of view, we have no experience with ERFE ELISA; on the other hand, we have a validated anti-ERFE antibody. Overall, the ERFE immunoblot data were included in order to confirm that low-iron diet indeed induces ERFE synthesis, as very strongly suggested by Fam132b PCR - both ELISA and immunoblotting can be expected to confirm the PCR results. As to serum hepcidin, again, we have no practical experience with hepcidin ELISA, but we are using validated primers for murine Hamp since 2004 and we believe there is a strong consensus that the regulation of murine Hamp is transcriptional. 

  1. Plasma iron levels are now added to the manuscript in Table 2.

Reviewer 2 Report

I have no additional comments

Author Response

We thank the Reviewer for her/his comments.

Round 3

Reviewer 1 Report

Dear Authors,

Given that an iron-deficient diet does not really work and does not alter the basic parameters of iron metabolism, especially in males, the question arises about generalizing conclusions for both sexes of mice. One should ponder the question why this is so. Is it the fault of the supplementation procedure or the fodder itself (from two sources), why? Unfortunately, I cannot find information about the supplementation time of the experiment and the age of the mice used in the experiment. From my experience I know that mice kept in cages with access to metal and even wood shavings or iron water, easily obtain iron even on a diet of 5ppm Fe.

It is alarming that in control mice, the levels of hemoglobin, plasma iron, and liver iron are unchanged following a low iron diet. Not even moderate anemia or hypoferremia was induced in control mice. If the diet does not work on the basic parameters equally for both sexes, it means that there are methodological problems. Perhaps, however, the authors' intention was not to induce anemia, but only some moderate state of iron deficiency? Can you comment on this and refer to the different effects of the diet on both sexes?

Obviously, the differences between the control animals in comparison to the mask mice are visible and clearly indicate the role of the spleen in the hematopoietic process and probably the faster iron turnover in this organ. It would be interesting to check the effect of an anemic diet in growing animals, where the iron stores from the liver and spleen would be used for growth and development.

Regards

Author Response

We thank the Reviewer for her/his comments. The age of C57 mice at the start of feeding (10 wk) and the duration of feeding (2 months) was by mistake only mentioned in the legend to Figure 6. (The experimental protocol for C57 mice was the same as for mask mice, which was correctly reported in the first sentence of Section 2.5). In the current third revision, the protocol is also explicitly included in the Materials and Methods section 4.1, as well as in the legend to Figure 4.

The fact that there is only a statistically insignificant drop in liver iron concentration and hemoglobin concentration in C57BL/6 mice on iron-deficient diet can be explained by the relatively high age of mice which, by the time of switching to iron-deficient diet, have already sufficiently high iron stores. It is well-known that feeding of iron deficient diet decreases iron parameters and hepcidin expression; however, most experiments reported in the literature use young growing mice, which have higher need for iron. Adult C57BL/6 mice are used less often and, in our experience, their response to iron deficient diet is much weaker and much more variable, which probably explains the lack of significant differences between control and iron-deficient C57BL/6 mice. When planning the iron deficient diet experiments reported in the manuscript, we indeed intended to induce only a mild state of iron deficiency in our mask mice, exactly as stated by the Reviewer. To this purpose, we choose to use adult animals. We certainly did not intend to induce severe iron deficiency anemia in mask mice, which already are iron-deficient, since there was a risk that the animals would not survive the treatment. The age of the C57 mice was then matched to the age of mask mice. In the third Revision, we included two sentences explaining this approach to the Materials and Methods section 4.1, as well as one additional reference (Reference 42).

As to the different effect on the two sexes, we can only speculate that, had the number of animals been higher, feeding of an iron deficient diet would have resulted in statistically significant differences in hepcidin expression, liver iron content and hemoglobin content in all groups. We have nevertheless reported the results for both sexes, as the observed tendencies in male mice support the results in female mice.  The difference in liver iron content between male and female C57 mice is well known; it might be that because of the higher baseline values in females, the effect of iron deficient diet is more pronounced. In any case, results from both genders support the main finding, namely the downregulation of hepcidin in iron-deficient mask mice.

Round 4

Reviewer 1 Report

The manuscript is much improved and the authors have responded to my comments.